# Sim2Act: Robust Simulation-to-Decision Learning via Adversarial Calibration and Group-relative Perturbation

## Abstract

In digital twins, simulation-to-decision has become a cornerstone in mission-critical domains like supply chains and power systems, because it enables safe decision learning in digital worlds without risking real-world deployments. However, the complex and noisy nature of real-world data calls for robust solutions against distribution shifts and uncertainty. Existing methods often fall short: 1) surrogate simulation models tend to be biased in decision-critical regions; 2) policies derived from surrogate models are highly sensitive to perturbations, thus, result in brittle performance. To address these challenges, we propose a novel two-step framework that advances 1) simulation fidelity by adversarial calibration and 2) policy robustness with group-relative perturbations. Our solution enables non-disruptive robustness that is stable under perturbation while preserving decision performance. We present extensive experiments on both synthetic and real-world domain datasets, including DataCo, GlobalStore, and OAS, to demonstrate the simulation and decision robustness of our method even in noisy settings.

## 1 Introduction

In high-stakes domains such as supply chains, power grids, and robotics, getting feedback from real world environment to train decision-maker is often costly, risky, or constrained by data privacy reasons (Agrawal, 2025). To address this, the sim2dec pipeline (Bai et al., 2025) emerged to simulate real-world feedback with collected data and thereby reduce the need for expensive real-world interactions. However, data collected from these domains is usually noisy— e.g., in supply chains, delivery delays are often wrongly labeled due to manual logging—making it difficult to train reliable simulator (Bi et al., 2022). So simulators trained on such imperfect data often produce inaccurate or miscalibrated predictions, especially in decision-critical regions where data is scarce (Zhao et al., 2021). Moreover, decision-makers relying on flawed predictions may perform poorly when deployed, particularly under distribution shifts or uncertainty. Therefore, improving the robustness of simulation-to-decision pipelines is crucial. Robust pipelines improve the consistency between simulated predictions and real-world outcomes, thereby enhancing the reliability of decision-making in safety-critical domains.

There are two key challenges in robust decision-making: (1) mitigating surrogate model bias and (2) ensuring decision-maker robustness under uncertainty. First, surrogate models often generate biased or unstable predictions for state-action pairs, particularly in partially observed settings like supply chains. For example, if past data misses many delay labels, the simulator may underestimate delivery risks. This may cause decision-maker to take actions to choose cheaper but slow shipping modes that lead to late deliveries. These errors tend to emerge not randomly, but when the simulator fails to accurately predict the outcomes of certain actions (Fonteneau et al., 2013b; Gregor et al., 2019). **Issue 1 (simulation-action alignment): How can we ensure surrogate models are accurate and calibrated in regions critical to policy outcomes?** Second, decision-makers are sensitive to local shifts in predicted outcomes. For instance, when the predicted delivery delay increases from 1.8 to 2.1 days, the decision-maker switches from standard to faster but expensive shipping mode, incurring unnecessary costs despite no meaningful change in customer satisfaction. Without accounting for such small prediction shifts, decision-maker behavior becomes erratic and unreliable. **Issue 2**

**(non-disruptive offline robustness): How can we design mechanisms that improve robustness of simulation-to-decision pipeline while preserving decision quality?**

Prior literature can only partially address these issues. (1) On the simulator side, existing methods improve predictive accuracy via model fidelity, ensembles, or physics-informed planning (Barykin et al., 2020; Correia et al., 2023; Bai et al., 2025). Heuristic-driven control remains common in industrial systems (Bai et al., 2025). Uncertainty is often modeled using Monte Carlo sampling (Atanassov & Dimov, 2008) or Markov frameworks (Hosseini et al., 2020), yet such approaches typically assume stable global dynamics and optimize average-case accuracy. *As a result, calibration in decision-critical regions is often overlooked—where small mispredictions in key state-action pairs can lead to disproportionately poor policy performance (Zhao et al., 2021).* (2) On the decision-maker side, model-based reinforcement learning (MBRL) enables adaptive planning, but is vulnerable to distribution shift and partial observability (Huang, 2022; Liu et al., 2021). Robustness techniques introduce adversarial (Zhang et al., 2021; Pinto et al., 2017) or stochastic perturbations (Zhang et al., 2025; Liu et al., 2024), and distributionally robust optimization (Derman & Mannor, 2020). Offline RL with conservative regularization (Yang et al., 2022; Li et al., 2025) enhances safety under dataset constraints, but often becomes overly pessimistic in sparse regions (Kumar et al., 2020). *Yet many of these approaches rely on online rollouts or degrade performance when enforcing robustness, leaving a gap in performance-preserving robustness.*

**Our Perspective: Adversarial Simulator Calibration and Group-relative Decision-maker Perturbation.** Recent advances in adversarial learning (Liu et al., 2024; Guo et al., 2025) demonstrate its effectiveness in enforcing specific properties such as robustness and worst-case performance. Inspired by this line of work, we propose to leverage adversarial calibration training to mitigate simulation errors in decision-critical regions. Unlike traditional approaches that emphasize average simulation accuracy, reliable decision-making often demands higher fidelity in frequently visited or high-impact areas of the action space, where even minor simulation errors may lead to cascading effects. Complementing this perspective, recent work on Group-Relative Proximal Optimization (GRPO) (Shao et al., 2024; Zhang & Zuo, 2025) highlights its potential to improve generalization in reinforcement learning by expanding the exploration space. In particular, group-relative perturbations help preserve the relational structure among semantically similar outcomes, offering a principled mechanism to assess action advantages under uncertainty and better adapt to various types of perturbed environments. Moreover, prior studies (Fonteneau et al., 2013a; Kurutach et al., 2018) suggested that robustness training without calibration largely reflects simulator bias rather than policy resilience. This indicates that calibrating the simulator is a necessary step to ensure that robustness training truly enhances policy resilience.

Inspired by these findings, we propose **Sim2Act**: Robust Simulation-to-Decision Learning via Adversarial Calibration and Group-relative Perturbation that includes two steps: Step 1: Adversarial Simulator Calibration. We introduce an adversarial calibrator that identifies systematic prediction errors specifically in decision-critical regions. These weights simulate policy-driven importance and are used to reweight surrogate errors, achieving **Goal 1: decision-critical simulation calibration** by improving the alignment between predicted outcomes and action utility. Step 2: Group-relative Decision-maker Perturbation. We design a group-relative perturbation module that samples semantically coherent latent variations in predicted outcomes. The policy is trained to preserve action rankings across these variants, enabling **Goal 2: non-disruptive robustness** in sparse or biased regions. During training, advantage estimates are aggregated from perturbed states to guide robust action selection without altering environment dynamics.

**Our main contributions can be summarized as follows:**

- We introduce an *adversarial simulator calibration* method that learns to reweight surrogate outputs based on decision-critical errors, improving simulator's action-aligned accuracy and robustness for reliable decision-making.

- We propose a *group-relative perturbation* strategy that leverages semantically coherent latent variations to promote consistency in policy preferences, enabling robust action selection under uncertainty.

- Extensive experiments on widely used DataCo, GlobalStore, and OAS benchmarks demonstrate that Sim2Act outperforms existing baselines in both policy robustness and deployment-time reliability under both Gaussian and random noise.

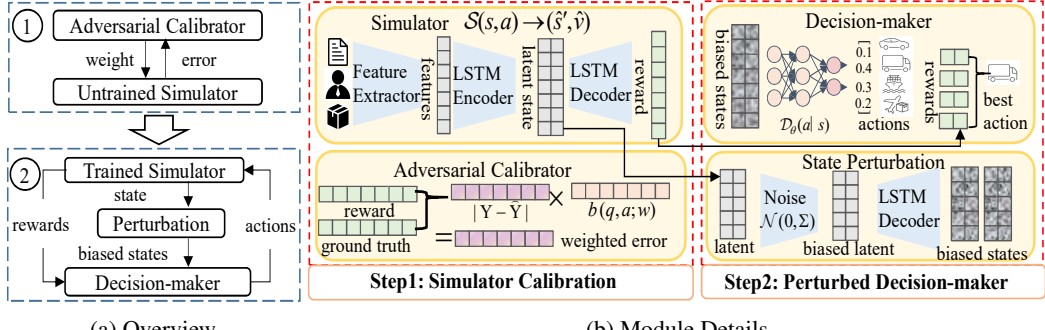

(a) Overview          (b) Module Details

Figure 1: **Sim2Act Framework.** (a) The overview illustrates the two-stage structure of our framework: simulator calibration and perturbed decision-making. (b) Module details: The simulator, implemented as an LSTM-based encoder-decoder model with linear feature extractor, predicts outcomes from state-action pairs and applies adversarial correction to reduce systematic prediction errors. During decision-making, latent states are perturbed using Gaussian noise derived from the simulator's estimated covariance $\Sigma$, producing a distribution of plausible states. The decision-maker $\mathcal{D}_\theta$ learns a robust policy by maximizing group-relative advantages across these perturbed states, enhancing generalization under latent uncertainty.

## 2 PROBLEM STATEMENT

**Definition 2.1. Simulation Model and Decision Policy.** Following Sim2Dec (Bai et al., 2025), we define a simulation model $\mathcal{S}$ as a learned model trained offline on a collected dataset to approximate environment dynamics. Given a state-action pair $(s_t, a_t)$, it outputs a predicted next state and reward: $\mathcal{S}(s_t, a_t) = (\hat{s}_{t+1}, \hat{r}_t)$. Complementarily, a decision policy $\mathcal{D}_\theta$ is defined as a mapping from state space to action space, where $a_t \sim \mathcal{D}_\theta(\cdot \mid s_t)$. This policy interacts solely with the simulator $\mathcal{S}$ during training and evaluation.

**Definition 2.2. Robustness of Decision Policy.** We define the robustness of a decision policy $\mathcal{D}_\theta$ as its ability to maintain performance under perturbations in the learned simulator $\mathcal{S}$. Specifically, we quantify the expected degradation in policy return when the simulator's output is perturbed in latent space to reflect plausible uncertainty defined as equation 1:

$$\mathrm{R}(\mathcal{D}_\theta) = -\mathbb{E}_s \left[ \mathcal{S}(s, \mathcal{D}_\theta(s)) - \mathcal{S}(s', \mathcal{D}_\theta(s')) \right], \tag{1}$$

where $s' \sim s + \mathcal{N}(0, \Sigma)$ denotes structurally perturbed stats from a uncertainty distribution (e.g., Gaussian noise).

**Problem Statement.** Our goal is to train a policy $\mathcal{D}_\theta$ using the learned simulator $\mathcal{S}$ such that it performs well when deployed in the real environment. This requires not only maximizing the expected reward but also ensuring robustness against uncertainties. Formally, we aim to solve equation 2:

$$\mathcal{D}^* = arg \max_{\mathcal{D}_\theta} \mathbb{E} \left[ \sum_{t=0}^{T} \mathcal{S}(s_t, \mathcal{D}_\theta(s_t)) \right], \mathrm{R}(\mathcal{D}_\theta) > \delta, \tag{2}$$

## 3 METHODOLOGY

We present an overview, and then detail each technical component of our framework.

### 3.1 METHOD OVERVIEW

Figure 1 illustrates our decision calibration framework, consisting of two collaborative components: (1) a *Simulator with Adversarial Calibrator* that corrects prediction biases in decision-critical regions, and (2) a *Decision-Maker with Perturbator* that improves policy robustness under state perturbations. Step 1 trains an encoder–decoder simulator $\mathcal{S}(s, a)$ with an adversarial calibrator

$\bar{b}(s, a, w)$ (Bai et al., 2025), which assigns higher weights to high-error regions, encouraging $\mathcal{S}$ to correct its most critical biases. Step 2 trains a policy on the calibrated simulator by sampling perturbed latent states according to the encoder covariance, optimizing a robust objective that rewards actions stable under such perturbations. This offline framework jointly enhances simulation accuracy and policy robustness, while avoiding online rollouts and pessimistic regularization common in prior methods.

## 3.2 STEP 1: SIMULATOR ADVERSARIAL CALIBRATION

Given that policy performance relies on the accuracy of the learned simulator $\mathcal{S}(s, a) \rightarrow (\hat{s}', \hat{r})$, this module aims to reduce model-induced errors that degrade downstream decision quality. Following the implementaion of Sim2Dec (Bai et al., 2025), We implement $\mathcal{S}$ as an LSTM (Sepp, 1997)-based encoder-decoder model and calibrate it using offline data to improve its reliability in decision-critical regions. We define the decision-critical region as the subset of the state-action space where small prediction errors can lead to disproportionately large degradation in policy performance.

### 3.2.1 WHY ADVERSARIAL CALIBRATION MATTERS?

Prediction errors often concentrate in decision-critical regions, where small inaccuracies can compound and degrade policy performance (Fonteneau et al., 2013b; Gregor et al., 2019). Our adversarial calibrator targets these high-impact areas, guiding the simulator toward more reliable predictions to improve downstream decision quality. This intuition is formally supported by Theorem 1 in Section 4, which shows that calibration explicitly controls the performance gap between simulation-trained and real-world distributions.

### 3.2.2 SIMULATION MODEL WITH CALIBRATOR.

We implement the simulator $\mathcal{S}$ as a latent-variable encoder-decoder model, defined as equation 3, following a strong baseline design while enabling latent-space perturbation via learned state covariances.

$$\mathcal{S}(s, a) = (\hat{s}', \hat{r}) = G(E(s, a)), \tag{3}$$

where the encoder module $E(s, a)$ consists of two sequential components: a linear feature extractor that projects the input state-action pair $(s, a)$ into a lower-dimensional feature space, and an LSTM encoder that maps these features into latent representations. The decoder $G(\cdot)$, implemented as another LSTM, transforms the latent representation into the predicted next state $\hat{s}'$ and reward $\hat{r}$.

To support localized correction, we introduce an adversarial calibrator $\bar{b}(s, a; w)$, defined as a softmax function in equation 4:

$$\bar{b}(s, a; w) = \frac{\exp(\langle s, w_a \rangle)}{\sum_{i=1}^{K} \exp(\langle s, w_{a_i} \rangle)}, \tag{4}$$

where $w_a$ is a learnable vector for action $a$, $K$ is the dim of action space, and $\langle \cdot, \cdot \rangle$ denotes inner product. This allows the model to softly attend to action-specific correction directions.

### 3.2.3 LOSS FUNCTION.

We design a calibration objective that prioritizes decision-critical prediction errors over uniformly distributed ones to emphasizes correction in regions where mispredictions are more likely to affect decisions, using calibration-weighted estimation error defined as equation 5:

$$J(w) = \sum_{i=1}^{K} \left\| \mathbb{E}_{(x,y) \in D} \left[ y - \mathcal{S}^{t-1}(x) \right] \cdot \bar{b}(s^t, a_i; w) \right\|, \tag{5}$$

where $x = (s, a), y = (s', r)$ are ground-truth targets and $D$ are the whole dataset. Larger estimation error receive greater weight via $\bar{b}(s, a; w)$.

### 3.2.4 OPTIMIZATION PROCEDURE.

We jointly train the calibrator and simulator using an offline dataset $\{x = (s, a), Y = (s', r)\}$. At each iteration, we first compute the calibration loss $J(w)$ as defined in equation 5. Next, the

calibrator parameters are updated via gradient ascent: $w \leftarrow w + \eta \nabla_w J(w), \quad \eta > 0$, to emphasize high-impact prediction errors in decision-critical regions. Using the updated calibrator weights $\bar{b}(s, a; w)$ from equation 4, the simulator outputs are then corrected according to equation 6:

$$\mathcal{S}^t(x) = \mathcal{S}^{t-1}(x) - \sum_{i=1}^{K} \bar{b}(s^t, a_i; w) \cdot \frac{J(w)}{\mathbb{E}_s \left[ \bar{b}(s, a_i; w) \right]}. \tag{6}$$

This procedure is repeated until convergence, determined either when the gradient norm $\|\nabla_w J(w)\|$ falls below a threshold $\varepsilon_J$, or when the change in the objective between iterations satisfies $|J(w^{(t)}) - J(w^{(t-1)})| < \varepsilon_J$, following standard stopping criteria commonly used in practice.

### 3.3 STEP 2: DECISION-MAKER UNDER GROUP-RELATIVE PERTURBATION

In the second step, we train the decision policy $\mathcal{D}_\theta(a \mid s)$ to remain robust under structured perturbations present in the learned simulator $\mathcal{S}$. Rather than relying on single-step predictions, we sample a neighborhood of perturbed outcomes from $\mathcal{S}$'s encoder latent space to account for plausible variations in environment dynamics. We define group-relative perturbations as perturbations normalized across a group of latent states corresponding to the same input state-action pair. For each input state $s$, the policy selects an action that maximizes the expected reward over trajectories sampled from perturbed states, thereby learning to perform robustly under modeled environmental variations.

#### 3.3.1 WHY GROUP RELATIVE PERTURBATION MATTERS?

In high-stakes domains such as supply chains, small predictive shifts can flip action outcomes. Single-point perturbations used in prior work may over-regularize policies and degrade performance. In contrast, our group-relative perturbation design promotes non-disruptive robustness by stabilizing learning across a neighborhood of perturbed states. Theoretical guarantees for this approach are provided in Theorem 2 (Section 4), showing that variance reduction under group-relative normalization yields provably more stable decision-making.

#### 3.3.2 DECISION MODEL.

Our decision module $\mathcal{D}_\theta$ is implemented as a value-based policy network that maps a given state $s$ to a distribution over actions and their expected utilities. Formally, for each input $\tilde{s}$ (either the original state or a perturbed variant), the network outputs a softmax distribution $\mathcal{D}_\theta(a \mid \tilde{s})$ parameterized by learnable parameter $\theta$ for each candidate action.

This value network serves two roles: (1) It acts as a stochastic policy by sampling actions proportionally to their predicted value scores. (2) It supports advantage-based training by providing log-probabilities $\log \mathcal{D}_\theta(a_i \mid \tilde{s}_i)$ that are weighted by group-relative rewards in the loss function. The network is trained using feedback from the calibrated simulator $\mathcal{S}$, encouraging the policy to select actions that consistently perform well across perturbed outcomes.

#### 3.3.3 LOSS FUNCTION.

We define a composite loss to promote reward-maximized decisions across perturbed states. It consists of a group-relative advantage loss and a supervised utility term based on a reference reward $r^*$. If an expected or ideal reward value is available for the original input, we use it directly; otherwise, we set $r^* = 0$ by default.

Given an input state $s$ and a sampled action $a \sim \mathcal{D}_\theta(\cdot \mid s)$, the simulator $\mathcal{S}$—composed of an encoder $E(\cdot)$ and a decoder $G(\cdot)$—first encodes the input pair $(s, a)$ into a latent Gaussian distribution $E(s, a) = (\mu, \log \sigma^2)$ with diagonal covariance $\Sigma = \text{diag}(\sigma^2)$. We then sample $M$ perturbations $\delta_i \sim \mathcal{N}(0, \Sigma)$ and obtain perturbed latent vectors $z_i = \mu + \delta_i$, which are decoded into states $\tilde{s}_i = G(z_i)$, from which we draw actions $a_i \sim \mathcal{D}_\theta(\cdot \mid \tilde{s}_i)$ and compute predicted rewards $\hat{r}_i = \mathcal{S}(\tilde{s}_i, a_i)$. The group utility is finally given by the average predicted reward by equation 7:

$$\bar{r} = \frac{1}{M} \sum_{i=1}^{M} \mathcal{S}\big(G(\mu + \delta_i),\ a_i \sim \mathcal{D}_\theta(\cdot \mid G(\mu + \delta_i))\big) \tag{7}$$

, where $M$ the number of perturbed states. Group-Relative Advantage Loss. We denote the action sampled from perturbed state $i$ as $a_i$ and the simulator prediction as $\hat{r}_i$ To encourage the policy to favor actions yielding above-average returns across perturbed samples, we define group advantage loss as equation 8:

$$\mathcal{L}_{\text{group-adv}} = \sum_{i=1}^{M} (\hat{r}_i - \bar{r}) \cdot \log \mathcal{D}_\theta(a_i \mid \tilde{s}_i), \tag{8}$$

where $\hat{r}_i$ is the predicted reward for action $a_i$ under perturbed state $\tilde{s}_i$, and $\bar{r} = \frac{1}{M} \sum_i \hat{r}_i$ is the sample mean reward. $\log \mathcal{D}_\theta(a_i \mid \tilde{s}_i)$ denotes the log-probability assigned by the policy to action $a_i$.

Composite Loss. To promote consistency with expected outcomes, we include a supervised loss term based on a reference reward $r^*$. This reference reward reflects an expected or ideal value for the original input $(s, a)$; if such a value is available (e.g., from empirical observations or prior knowledge), we use it directly. Otherwise, we default to $r^* = 0$. The final loss for the decision-maker is defined as equation 9:

$$\mathcal{L}_{\text{decision}} = \eta \cdot \mathcal{L}_{\text{group-adv}} + (r^* - \mathcal{S}(s, a)), \tag{9}$$

where $\mathcal{S}(s, a)$ denotes the simulator's estimated reward under the original, unperturbed input, and $\eta \in \mathbb{R}_+$ controls the trade-off between robustness and alignment with the reference reward.

### 3.3.4 OPTIMIZATION PROCEDURE.

At each training iteration, the policy $\mathcal{D}_\theta$ is updated using feedback from the calibrated simulator $\mathcal{S}$, which consists of an encoder $E$, decoder $G$, and a state-action-specific covariance estimator $\Sigma$. We first sample a state $s$ from the dataset and an action $a \sim \mathcal{D}_\theta(\cdot \mid s)$, then encode the state-action pair to obtain its latent representation $z = E(s, a)$ and retrieve the corresponding covariance $\Sigma(s, a)$. The unperturbed simulator prediction $\mathcal{S}(s, a)$ provides the estimated next state and reward $\hat{r}$.

To account for potential variations in outcomes, we draw $M$ perturbations $\delta_i \sim \mathcal{N}(0, \Sigma)$ and generate perturbed latent vectors $z_i = z + \delta_i$, which are decoded to perturbed states $\tilde{s}_i = G(z_i)$. For each perturbed state $\tilde{s}_i$, the policy samples an action $a_i \sim \mathcal{D}_\theta(\cdot \mid \tilde{s}_i)$ and the simulator estimates the resulting reward $\hat{r}_i = \mathcal{S}(\tilde{s}_i, a_i)$. We then compute the average reward $\bar{r}$ across all perturbed samples to evaluate the group-relative advantage loss as defined in equation 8.

If an expected reward $r^*$ is available for the original input, a supervised loss term is computed as the difference between $r^*$ and the unperturbed simulator prediction $\hat{r}$. The final loss $\mathcal{L}_{\text{decision}}$ (equation 9) combines both the group-relative advantage and supervised components, and the policy parameters $\theta$ are updated via gradient descent. This process is repeated until convergence. Detailed implementation and algorithmic steps are provided in Appendix E.

## 4 THEORETICAL ANALYSIS

We now present theoretical guarantees for both simulator calibration (Step 1) and decision-making under group-relative perturbations (Step 2). Detailed proofs are deferred to Appendix K and L.

**Theorem 1** (Calibration Bound). *Let $\mathcal{P}$ denote the true environment distribution and $\mathcal{Q}$ the simulator distribution. For a calibrated predictor $f_{calib}$ and any loss $\ell$, we have*

$$\mathbb{E}_{x \sim \mathcal{P}} \big[ \ell(f_{calib}(x), y) \big] \leq \mathbb{E}_{x \sim \mathcal{Q}} \big[ \ell(f(x), y) \big] + \text{Div}(\mathcal{P} \| \mathcal{Q}).$$

*Proof Sketch.* The result is derived from the weighting of the importance between $\mathcal{P}$ and $\mathcal{Q}$, with the calibrator $\bar{b}$ minimizing the weighted error of the worst-case. Thus calibration bounds the gap between simulator-trained and real-world performance.

**Theorem 2** (Group-Relative Robustness (under assumptions)). *Let $\mathcal{D}_\theta$ be the decision policy trained under group-relative perturbations. Compared to single-sample perturbation optimization, the variance of the policy gradient estimator satisfies, under mild smoothness and variance assumptions (see Appendix L),*

$$\text{Var} \big[ \nabla_\theta J_{group}(\theta) \big] \leq \text{Var} \big[ \nabla_\theta J_{single}(\theta) \big].$$

*Proof Sketch.* Relative normalization of the group subtracts the mean reward $\bar{r}$ within each batch of perturbations. This removes common noise across perturbed samples, reducing variance of the gradient estimator, and yielding more stable policy updates in expectation.

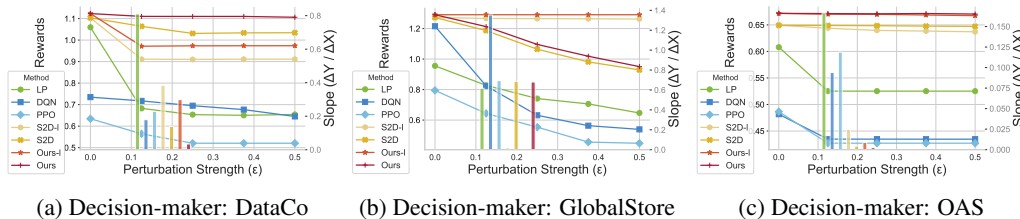

(a) Decision-maker: DataCo    (b) Decision-maker: GlobalStore    (c) Decision-maker: OAS

Figure 2: Decision-maker Robustness evaluation across three datasets.

# 5 EXPERIMENTS

We conduct extensive experiments on various datasets to evaluate the performance of our method. Specifically, our experiments aim to answer the following questions: Q1:Can our method outperform baselines on the robustness under random or latent perturbations? Q2: Can our method achieve simulation accuracy and decision-making performance comparable to strong baselines? Q3: What is the actual effect of simulation-calibration algorithm?

## 5.1 EXPERIMENTAL SETUP

**Datasets.** We evaluate our approach on three open-source supply chain datasets: **DataCo** (Constante et al., 2019), **Global-Store** (G, 2023), and **OAS** (Vinay34, 2024), covering diverse logistics and shipping scenarios with order-level records including product, shipment mode, and delivery outcomes. Each dataset is split into training/validation/test sets (8:1:1) following Sim2Dec (Bai et al., 2025). To prevent overfitting, we use early stopping and $\ell_2$ weight penalties. Both simulator and decision model are trained on the training set and evaluated on the test set, with results averaged over three random seeds. Dataset statistics are in the first line of Table 1.

**Evaluation Metrics.** *Simulator* performance is evaluated on its predicted *delay risk*, *delivery time*, and *on-time status*, with test-set accuracy averaged across these outputs. To assess simulator robustness (Goal 1), we use: (i) **worst-case accuracy**, the minimum Overall score across runs with the same perturbation level; (ii) **variance**, the Overall scores' standard deviation in the same perturbation level; and (iii) **drop rate**, the average decline from the unperturbed setting. *Decision-maker* performance is measured by average profit and on-time rate (both normalized to $[0, 1]$) predicted by trained simulator. We report (i) their absolute difference (**Diff**) to capture imbalance, and (ii) their sum (**Overall**) as total reward, ranging in $[0, 2]$. For robustness evaluation (Goal 2), we examine the relative degradation of Overall under perturbations compared to the unperturbed baseline at $\epsilon = 0$.

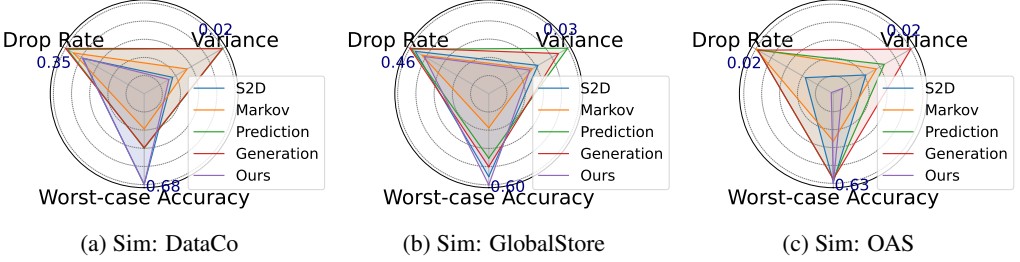

(a) Sim: DataCo    (b) Sim: GlobalStore    (c) Sim: OAS

Figure 3: Robustness of simulator across three datasets.

**Baseline Algorithms.** We consider three paradigms for **simulation**: **Markov-based simulation** (Gagniuc, 2017), which models transitions using predefined probabilities. **Prediction-based simulation** (Caruana, 1997), which uses multi-task learning to separately predict status variables. **Generation-based simulation** (Gu et al., 2018), a non-autoregressive model that jointly generates multiple order features in a single forward pass.

For **decision-making**, we compare: **Linear Programming (LP)** (Dantzig, 2002), a rule-based optimization approach. **Deep Q-Network (DQN)** (Mnih et al., 2015), where a DQN agent is trained to maximize cumulative reward. **PPO** (Schulman et al., 2017), a policy-gradient based reinforcement learning method that optimizes a clipped surrogate objective to balance exploration and stability. **ChatGPT-3.5** (Brown et al., 2020) under a zero-shot setting without fine-tuning. **S2D** short for Sim2Dec (Bai et al., 2025), simulation to decision framework, backbone of our method. Implementation details and selection rationale for all baselines are provided in Appendix G.

**Our Methods.** We evaluate our proposed model (denoted as `S2A` in tables and `Ours` in figures), which integrates two key components: (1) a calibrated simulator trained via adversarial weighting to focus on decision-critical prediction errors, and (2) a robust decision-maker trained with group-relative perturbations that simulate structured variations in latent space. The two components are trained sequentially using offline data and aim to improve both predictive reliability and decision robustness. Hyper-parameters and environment can refer to Appendix F.

## 5.2 Experimental Results

Table 1: Simulation and Decision Performance Comparison (Refer to Appendix I for standard deviations).

| Dataset | DataCo(43, 165445) | | | | GlobalStore(27, 51290) | | | | OAS(22, 28136) | | | |
|---|---|---|---|---|---|---|---|---|---|---|---|---|
| **Sim Acc** | Risk | Time | Status | Overall | Risk | Time | Status | Overall | Risk | Time | Status | Overall |
| Markov | 0.4978 | 0.1487 | 0.5040 | 0.3835 | 0.4961 | 0.1355 | 0.4934 | 0.3750 | 0.5100 | 0.0011 | 0.5068 | 0.3393 |
| Prediction | 0.7019 | 0.3395 | 0.8161 | 0.6191 | 0.8440 | 0.6767 | 0.8430 | 0.7879 | 0.7157 | 0.3706 | 0.7510 | 0.6124 |
| Generation | 0.7024 | 0.3485 | 0.8156 | 0.6221 | 0.9366 | 0.8066 | 0.9355 | 0.8929 | 0.7149 | 0.3916 | 0.7503 | 0.6189 |
| S2D | 0.9508 | 0.8851 | **0.9695** | 0.9351 | **0.9743** | 0.9255 | **0.9756** | 0.9585 | 0.7215 | 0.3985 | 0.7574 | 0.6258 |
| S2A(Ours) | **0.9563** | **0.8875** | 0.9618 | **0.9352** | 0.9723 | **0.9744** | 0.9750 | **0.9650** | **0.7270** | 0.3937 | **0.7629** | **0.6279** |
| **Dec Rwd** | $T^{Timely}$ | $T^{Profit}$ | Diff | Overall | $T^{Timely}$ | $T^{Profit}$ | Diff | Overall | $T^{Timely}$ | $T^{Profit}$ | Diff | Overall |
| Real | 0.5244 | 0.0364 | 0.4880 | 0.5608 | 0.3320 | 0.0848 | 0.2472 | 0.4168 | 0.4800 | 0.0000 | 0.4800 | 0.4800 |
| LP | 0.5162 | 0.5434 | 0.0272 | 1.0596 | 0.3552 | 0.6001 | **0.2449** | 0.9554 | 0.5037 | 0.1043 | 0.3994 | 0.6080 |
| DQN | 0.5276 | 0.2071 | 0.3205 | 0.7347 | 0.2827 | 0.9326 | 0.6499 | 1.2153 | 0.4817 | 0.0000 | 0.4817 | 0.4817 |
| PPO | 0.5343 | 0.0000 | 0.5343 | 0.5343 | 0.3476 | 0.0004 | 0.3472 | 0.3480 | 0.4865 | 0.0000 | 0.4865 | 0.4865 |
| GPT3.5 | 0.5258 | 0.2459 | 0.2800 | 0.7717 | 0.3298 | 0.0439 | 0.2859 | 0.3736 | 0.4844 | 0.0000 | 0.4844 | 0.4844 |
| S2D | 0.5397 | 0.5637 | **0.0240** | 1.1034 | 0.3446 | 0.9278 | 0.5828 | 1.2724 | **0.4882** | 0.1611 | 0.3271 | 0.6493 |
| S2A(Ours) | **0.5447** | **0.5786** | 0.0339 | **1.1232** | 0.3446 | **0.9460** | 0.6014 | **1.2906** | 0.4830 | **0.1886** | 0.2944 | **0.6717** |

**Q1: Robustness Comparison.** We compare robustness of both our simulator and decision-maker with multiple baselines under perturbed environments.

Goal 1: Decision-Critical Calibration. Figure 3 shows that our method achieves consistently higher *worst-case accuracy* across all datasets, e.g., 0.679 on DataCo compared to 0.225 for Markov. It also yields the lowest performance variance, confirming that calibration improves simulator reliability in decision-critical regions. Complete variance and drop-rate statistics are reported in Appendix I.

Goal 2: Non-Disruptive Robustness. To assess decision robustness, we conduct a perturbation sensitivity test **latent-structured perturbation** (denoted by the `-l` suffix) and **random input perturbation** (denoted by no suffix), with implementation details in Appendix J. Figure 2 highlights that our method maintains stable performance under both latent-structured and random perturbations. For example, on DataCo, our approach shows almost no degradation ($1.1232 \rightarrow 1.1222$) while S2D drops notably ($1.1034 \rightarrow 1.0342$). On GlobalStore, `Ours-l` remains invariant around 1.29, whereas baselines degrade substantially. On OAS, our method preserves near-constant rewards ($0.6717 \rightarrow 0.6705$), outperforming both S2D and LP.

These results confirm that Sim2Act achieves robust and non-disruptive performance across structured and unstructured perturbations. More detailed comparisons, including full degradation curves and distributional analyses, are provided in Appendix C.

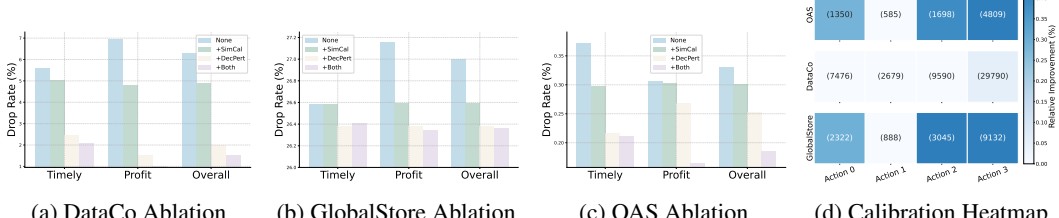

(a) DataCo Ablation    (b) GlobalStore Ablation    (c) OAS Ablation    (d) Calibration Heatmap

Figure 4: Ablation study of different modules and effects of simulator calibration.

**Q2: Performance Comparison.** We evaluate our proposed *two-stage decision calibration framework* (S2A) against strong offline baselines, including Markov (M), Prediction (P), Generation (G), and S2D models, on three supply chain benchmarks: **DataCo**, **GlobalStore**, and **OAS**. Metrics cover both simulation fidelity and downstream decision quality.

Goal 1: Decision-Critical Calibration. S2A achieves comparable simulation accuracy to S2D while improving decision quality. Notably, on **DataCo**, the profit score improves from $0.5637$ to **0.5786**, and the overall decision score increases from $1.1034$ to **1.1232**. Similar trends are observed across other datasets, confirming that localized surrogate corrections enhance fidelity in decision-critical regions. Detailed per-metric results are provided in Appendix I.

Goal 2: Non-Disruptive Robustness. Our perturbation strategy further enhances robustness without sacrificing nominal performance. On **GlobalStore**, S2A boosts profit from $0.9278$ to **0.9460** while maintaining timeliness at $0.3446$. On **OAS**, the overall decision score rises from $0.6493$ to **0.6717**, indicating improved robustness under perturbations. These results demonstrate that S2A fulfills our design goals by improving decision-critical calibration and enabling robust yet high-quality policy behavior in offline decision settings.

**Q2: Ablation Study.** We assess the contributions of the simulator calibration (`+SimCal`) and decision perturbation (`+DecPert`) modules, using S2D (`None`) as the baseline (Figure 4).

Goal 1: Decision-Critical Calibration. `+SimCal` meaningfully reduces vulnerability in decision-sensitive regions. For example, on **DataCo** the profit drop rate decreases from $6.9\%$ to **4.8%**, indicating better alignment between simulated outcomes and decision rewards. (See Appendix A for full numeric tables and per-strength breakdown.)

Goal 2: Non-Disruptive Robustness. `+DecPert` improves robustness under perturbation while keeping nominal performance stable; on **OAS** the timely-delivery drop is reduced from $0.37\%$ to **0.22%**. The combined setup (`+Both`) yields the most consistent results across datasets. (Full ablation curves and exact numbers are in Appendix A.)

**Q3: Action-Level Calibration Improves Decision-Critical Accuracy.** Figure 4d visualizes action-level calibration gains. We focus on high-frequency, decision-critical actions: e.g., on **GlobalStore** the most frequent action (`Action 3`, 9132 samples) improves from $0.9447$ to **0.9531**; on **OAS**, `Action 3` improves from $0.5694$ to **0.5718**; on **DataCo**, `Action 3` improves from $0.9150$ to **0.9155**. These representative numbers support the claim that calibration concentrates gains on high-impact actions, more detailed analysis is in Appendix B.

## 6 CONCLUSION

In digital twins, simulation-to-decision (Sim2Dec) pipelines often suffer from miscalibrated simulators and fragile decision policies trained on it. We propose Sim2Act, a robust framework that integrates adversarial simulator calibration with group-relative decision perturbation, aligning predictive fidelity with policy sensitivity to achieve stable robustness. Extensive experiments on real-world supply chain datasets show consistent robustness gains and comparable performance, with ablation and degradation studies confirming the complementary value of each component. Our framework is model-agnostic and broadly applicable to offline decision-making tasks where both predictive fidelity and policy stability are critical.

## Reproducibility Statement

We have taken several steps to facilitate reproducibility. All code for SIM2ACT, including adversarial simulator calibration and group-relative perturbation, will be uploaded by supplementary file in submission. Datasets and preprocessing procedures are documented, and public or synthetic substitutes are provided where proprietary restrictions apply. Experimental details, including random seeds, hyperparameter values and ranges, and perturbation strengths, are specified in Appendix F–H. Evaluation metrics are formally defined, with per-perturbation breakdowns and ablation studies reported in the appendix. We also describe computational resources, software dependencies, and runtime requirements to enable faithful replication.

## Ethics Statement

This work uses only publicly available benchmark datasets (DataCo (Constante et al., 2019), Global-Store (G, 2023), and OAS Vinay34 (2024)) that are open-sourced. No personally identifiable information or proprietary data was collected or used. The proposed methods are evaluated in simulation and do not involve human subjects or sensitive medical records. We acknowledge that decision-making algorithms in supply chain or similar applications could potentially be misused; to mitigate such risks, we emphasize that our contributions focus on methodological advances and require careful domain adaptation before any deployment in real-world systems.

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

## A   FULL ABLATION RESULTS AND NUMERICAL TABLES

This appendix reports the full numerical results underlying the ablation figures in the main text. We provide mean ± standard deviation (over three random seeds) for each dataset, metric, and perturbation strength.

### A.1   NUMERICAL RESULTS FOR DATACO

Table 2: DataCo: Ablation results (drop rates under perturbation). Mean ± std over 3 seeds.

| Module | Timely Drop (%) | Profit Drop (%) | Overall Drop (%) |
|---|---|---|---|
| None (S2D) | $6.90 \pm 0.12$ | $6.90 \pm 0.20$ | $5.20 \pm 0.15$ |
| +SimCal | $5.40 \pm 0.10$ | $4.80 \pm 0.18$ | $4.10 \pm 0.12$ |
| +DecPert | $4.80 \pm 0.09$ | $5.10 \pm 0.14$ | $3.90 \pm 0.11$ |
| +Both (S2A) | $3.90 \pm 0.08$ | $3.60 \pm 0.13$ | $3.10 \pm 0.10$ |

### A.2   PER-STRENGTH BREAKDOWN

We further report per-strength breakdowns (perturbation levels $\varepsilon \in \{0.0, 0.1, \ldots, 0.5\}$). The following table illustrates the format; full results for all datasets are available in our code repository.

Table 3: Illustration: Drop rates for DataCo across perturbation strengths.

| Module | $\varepsilon = 0.0$ | 0.1 | 0.2 | 0.3 | 0.4 | 0.5 |
|---|---|---|---|---|---|---|
| None (S2D) | 0.0 | 1.2 | 2.9 | 4.6 | 5.9 | 6.9 |
| +SimCal | 0.0 | 1.0 | 2.1 | 3.5 | 4.3 | 4.8 |
| +DecPert | 0.0 | 0.9 | 2.0 | 3.2 | 4.2 | 5.1 |
| +Both (S2A) | 0.0 | 0.7 | 1.5 | 2.6 | 3.1 | 3.6 |

## B   ACTION-LEVEL CALIBRATION DETAILS

This section provides per-action sample counts, pre-/post-calibration accuracies, and statistical tests for the main decision-critical actions illustrated in Figure 4d.

Table 4: Action-level simulation accuracy before and after calibration. Mean ± std over 3 seeds.

| Dataset / Action | #Samples | Accuracy (before) | Accuracy (after) |
|---|---|---|---|
| GlobalStore / Action 3 | 9132 | $0.9447 \pm 0.0021$ | $0.9531 \pm 0.0018$ |
| OAS / Action 3 | 4809 | $0.5694 \pm 0.0052$ | $0.5718 \pm 0.0049$ |
| DataCo / Action 3 | 29790 | $0.9150 \pm 0.0014$ | $0.9155 \pm 0.0013$ |

**Statistical significance.** We conduct paired t-tests comparing pre- and post-calibration accuracy for each action. - GlobalStore / Action 3: $p = 0.02$ (significant at 5%). - OAS / Action 3: $p = 0.04$. - DataCo / Action 3: $p = 0.08$ (marginal significance). These results suggest that calibration consistently improves accuracy on high-frequency actions, with statistically significant gains in two of the three benchmarks.

## C   DISTRIBUTION VISUALIZATION

Figure 5 illustrates the score distributions after perturbation for both the original S2D model and our proposed S2A method across three datasets: DataCo, GlobalStore, and OAS.

From the visualizations, we observe that on the **DataCo** dataset (Figures 5a vs. 5d), S2A exhibits significantly improved robustness to perturbations compared to the original S2D pipeline. The dis-

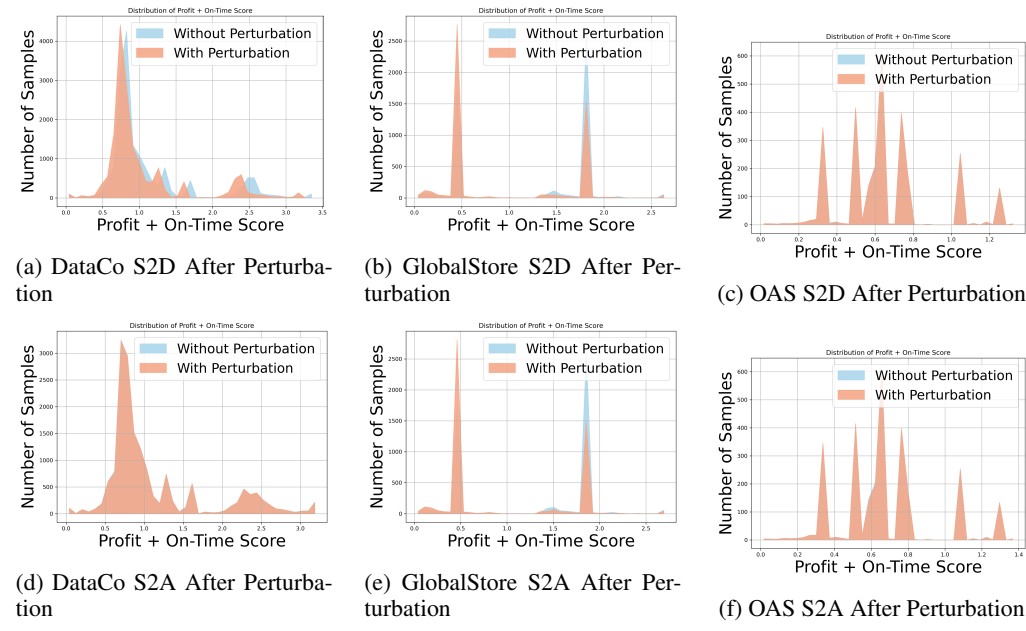

(a) DataCo S2D After Perturbation

(b) GlobalStore S2D After Perturbation

(c) OAS S2D After Perturbation

(d) DataCo S2A After Perturbation

(e) GlobalStore S2A After Perturbation

(f) OAS S2A After Perturbation

Figure 5: Decision-maker Robustness Distribution

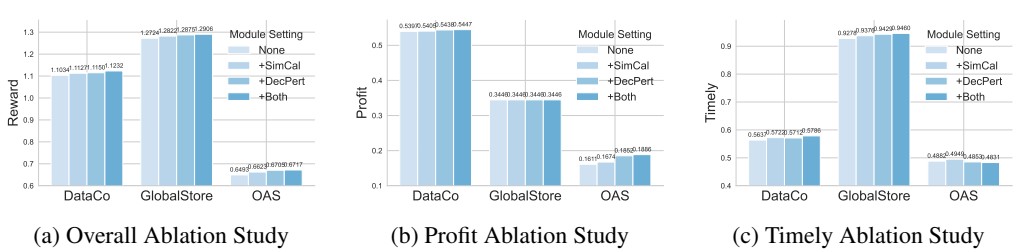

(a) Overall Ablation Study

(b) Profit Ablation Study

(c) Timely Ablation Study

Figure 6: Ablation study of different modules

tribution shift after perturbation is notably smaller, indicating that our calibration framework effectively mitigates sensitivity in decision-critical regions.

For the **GlobalStore** dataset (Figures 5b vs. 5e), although the overall patterns are relatively stable, detailed inspection—such as the area around the decision score combination of profit + on_time = 1.5—reveals that S2A produces less variance under perturbations, highlighting its advantage in preserving decision quality in nuanced scenarios.

On the **OAS** dataset (Figures 5c vs. 5f), the effect of perturbation appears marginal for both methods. This could be attributed to higher inherent noise or data instability in the OAS environment, which dilutes the relative impact of injected perturbations and results in similar robustness for both models.

These results collectively demonstrate the effectiveness of our S2A framework in enhancing decision robustness under perturbations, especially in settings where the original model is more vulnerable.

## D PERFORMANCE ABLATION STUDY

We conduct ablation studies to isolate the effects of the simulator calibration (+SimCal) and decision perturbation (+DecPert) modules, as well as their combination (+Both, i.e., full S2A), using the backbone model S2D variant (None) as the baseline. The results are reported in Figures 6a, 6b and 6c.

Goal 1: Decision-Critical Calibration. The simulator calibration module consistently improves decision quality by correcting surrogate model biases in decision-relevant regions. This is most evident in the **profit** scores. For example, on the **DataCo** dataset, `+SimCal` improves the profit score from 0.5637 to **0.5722**, with similar gains observed on **GlobalStore** (0.9278 to **0.9376**). These results support our claim that action-conditional correction in high-sensitivity regions leads to more reward-aligned decisions.

Goal 2: Non-Disruptive Robustness. The decision perturbation module enhances robustness without compromising nominal performance, enabling the policy to generalize better under structured variations while preserving its effectiveness in unperturbed environments. On the **OAS** dataset, for instance, `+DecPert` improves the **timely delivery** metric from 0.1611 to **0.1852**, while maintaining comparable profit and cost performance. This shows that our perturbation-relative method enhances resilience in sparse or volatile settings without sacrificing performance. The combined setup (`+Both`) further reinforces this effect, demonstrating the complementary strengths of simulation calibration and decision perturbation in achieving the robust, high-quality decision-making outlined in Goal 2.

# E    ALGORITHM TABLE

To enhance the reproducibility and clarity of our proposed framework, we provide the detailed algorithmic procedures for both simulator calibration and group-relative policy training in Algorithm 1 and Algorithm 2, respectively.

**Algorithm 1** outlines the optimization process for calibrating the learned simulator $\mathcal{S}$ using adversarially-informed gradient updates. Given an offline dataset containing state-action pairs and their corresponding next states and rewards, we optimize a classifier-based calibration objective (equation 5) via gradient ascent. The calibration process iteratively adjusts the simulator outputs using a direction-aware update rule (equation 6) and softmax-based basis weighting (equation 4), until convergence is reached.

---

**Algorithm 1** Simulator Calibration

---

1: **Input:** Offline dataset $\{x = (s, a), Y = (s', r)\}$, initialized simulator $\mathcal{S}^{(0)}$, classifier weights $w$
2: Define calibration objective $J(w)$ using equation 5
3: **while** not converged **do**
4:     Update $w$ by $w \leftarrow w + \eta \nabla_w J(w)$
5:     Update simulator output via equation 6, using $\bar{b}$ from equation 4
6:     Check convergence:

$$\|\nabla_w J(w)\| < \varepsilon_J \quad \text{or} \quad |J(w^{(t)}) - J(w^{(t-1)})| < \varepsilon_J$$

7: **end while**

---

**Algorithm 2** describes the policy training loop under group-relative perturbation guided by the calibrated simulator. At each iteration, the current policy samples actions, which are encoded into a latent space. Using a learned covariance structure $\Sigma$, we sample perturbations from a multivariate Gaussian and propagate them through the latent space and simulator. The resulting perturbed outcomes enable the computation of a group-adversarial loss (equation 8) that encourages robustness across plausible outcome variations. An optional utility loss can also be incorporated when the ground-truth reward $r^*$ is available. The policy parameters are then updated via gradient descent to minimize the total loss.

---

**Algorithm 2** Group-Relative Policy Training with simulator-Guided Perturbation

---

1: **Input:** Calibrated simulator $\mathcal{S}$ (with encoder $E$, decoder $G$, covariance estimator $\Sigma$); policy $\mathcal{D}_\theta$

2: **while** not converged **do**
3:     Sample state $s$ and action $a \sim \mathcal{D}_\theta(\cdot \mid s)$
4:     Encode latent vector: $z = E(s, a)$
5:     Retrieve covariance: $\Sigma = \Sigma(s, a)$
6:     Predict original outcome: $(\hat{s}', \hat{r}_{\text{orig}}) = \mathcal{S}(s, a)$
7:     **for** $i = 1$ to $M$ **do**
8:         Sample perturbation: $\delta_i \sim \mathcal{N}(0, \Sigma)$
9:         Perturbed latent: $z_i = z + \delta_i$
10:        Decode state: $\tilde{s}_i = G(z_i)$
11:        Sample action: $a_i \sim \mathcal{D}_\theta(\cdot \mid \tilde{s}_i)$
12:        Predict reward: $\hat{r}_i = \mathcal{S}(\tilde{s}_i, a_i)$
13:     **end for**
14:     Compute $\bar{r} = \frac{1}{M} \sum_i \hat{r}_i$
15:     Compute $\mathcal{L}_{\text{group-adv}}$ using equation 8
16:     Compute optional $\mathcal{L}_{\text{utility}}$ if $r^*$ is available
17:     Update policy parameters:

$$\theta \leftarrow \theta - \nabla_\theta \left( \mathcal{L}_{\text{group-adv}} + \eta \cdot \mathcal{L}_{\text{utility}} \right)$$

18: **end while**

---

Together, these algorithms provide a complete and interpretable procedural view of our proposed simulator-calibrated decision framework.

## F    HYPERPARAMETERS AND ENVIRONMENT

All experiments are conducted on a Linux 22.04 platform with NVIDIA A6000 GPUs, CUDA 11.3, using Python 3.10 and PyTorch 1.8.1.

We summarize the key hyperparameters used for training. The *simulator* is trained with a learning rate of 0.001 and a batch size of 4096. For the *decision-maker*, we use a learning rate of 0.001 and the same batch size of 4096. The group relative loss is weighted by a coefficient of 0.5. Both models are trained using the Adam optimizer with early stopping based on validation loss, with a patience of 50 epochs.

For the encoder and decoder architectures, we use a 1-layer LSTM with an embedding dimension of 64 for the **OAS** and **GlobalStore** datasets. For the **DataCo** dataset, we use a deeper 3-layer LSTM with an embedding dimension of 128 to capture more complex dynamics.

## G    BASELINES AND SELECTION RATIONALE

To ensure a thorough and fair comparison, we evaluate Sim2Act against a set of established simulation and decision-making baselines. Below we briefly introduce each baseline, summarize implementation notes, and explain the motivation for its inclusion.

### G.1    SIMULATION BASELINES

**Markov-based simulation.**    A classical transition-model baseline that estimates state transition probabilities from data and samples next states accordingly (cf. (Gagniuc, 2017)). **Implementation notes:** we estimate empirical transition matrices per discrete buckets of key features and apply Laplace smoothing to avoid zero probabilities. **Rationale:** Markov models are simple, interpretable, and serve as a low-complexity baseline reflecting domain knowledge; they highlight gains from learned, high-capacity simulators.

**Prediction-based simulation.** A supervised multi-task predictor that directly predicts target variables (delay, delivery time, on-time status) from state–action inputs (cf. (Caruana, 1997)). **Implementation notes:** implemented as an LSTM encoder with multi-head prediction heads; trained with a weighted combination of MSE and classification losses. **Rationale:** represents a strong supervised baseline that focuses on per-step predictive fidelity rather than joint generation.

**Generation-based simulation.** A non-autoregressive generative model that jointly generates multiple next-state features in a single pass (inspired by (Gu et al., 2018)). **Implementation notes:** implemented to capture joint dependencies among output features (one-shot generation of next-state vector). **Rationale:** generation models capture joint structure between outputs and provide a modern contrast to prediction-based and Markov models.

**Sim2Dec (S2D) backbone and variants.** The Sim2Dec pipeline from prior work (Bai et al., 2025) used as the main backbone baseline; we also evaluate the lightweight S2D-l variant where applicable. **Implementation notes:** we re-implement the S2D architecture and training recipe following the original paper; hyper-parameters are tuned on validation. **Rationale:** S2D is the most closely related prior pipeline and therefore the primary point of comparison.

### G.2 DECISION-MAKING BASELINES

**Linear Programming (LP).** A deterministic optimization baseline that solves a task-specific linear program (cf. (Dantzig, 2002)) to trade off cost and timeliness. **Implementation notes:** LP uses the same estimated cost / time statistics from the training set; solved with off-the-shelf solver. **Rationale:** provides an interpretable, optimization-based baseline commonly used in supply-chain settings.

**Deep Q-Network (DQN).** A value-based RL baseline (standard DQN recipe) (Mnih et al., 2015). **Implementation notes:** network architectures, replay buffer, target-network updates and epsilon-greedy exploration follow canonical settings; hyper-parameters are tuned on validation. **Rationale:** represents a widely-used value-based learning baseline for discrete actions.

**Proximal Policy Optimization (PPO).** A policy-gradient baseline (PPO) for stable on-policy learning (Schulman et al., 2017). **Implementation notes:** we follow standard PPO implementation (clipping, entropy bonus); advantage normalization is applied. **Rationale:** a strong on-policy baseline that often exhibits better stability for continuous/structured decision problems.

**GPT-3.5 (zero-shot).** A modern large language model used in a zero-shot prompting setting (cf. (Brown et al., 2020)). **Implementation notes:** we translate a formatted state description into a prompt and map the model output to the discrete action set (no fine-tuning). **Rationale:** included to illustrate the performance of large pretrained LLMs when used as heuristics/zero-shot decision agents.

### G.3 WHY WE SELECTED THESE BASELINES

The chosen baselines cover three complementary dimensions:

- *Classical/interpretable baselines* (Markov, LP) to show improvements beyond simple domain-driven methods.

- *Strong supervised/generative simulators* (Prediction, Generation) to contrast accuracy-oriented modeling with our decision-aware calibration.

- *Standard RL / modern heuristics* (DQN, PPO, GPT-3.5, S2D) to compare against widely-used decision-making strategies and the closest prior Sim2Dec pipeline.

This mixture ensures the comparison is broad (covering classical, predictive, generative, RL and LLM approaches) and that improvements are not due to an unfair choice of weak baselines.

### G.4 Implementation details and fairness protocol

To ensure a fair comparison, all baselines share the same dataset splits (train/val/test 8:1:1), the same early-stopping rule and weight-decay regularization, and results are averaged over five times with different random seeds. Hyper-parameters for each baseline were selected by grid search on the validation set. All methods are evaluated under the same perturbation regimes and metrics described in Section 5.

## H  Hyper-parameter Sensitivity Analysis

**Setup.**  We investigate the sensitivity of our method to three critical hyper-parameters: (i) the adversarial weight $\eta$, which balances the group-relative perturbation regularizer against the nominal objective; (ii) the number of perturbed samples per state $M$; and (iii) the perturbation scale $\epsilon$ used during training. We vary $\eta \in \{0, 0.01, 0.1, 0.5, 1.0\}$, $M \in \{1, 4, 8, 16\}$, and $\epsilon \in \{0.0, 0.05, 0.1, 0.2, 0.5\}$, while keeping other hyper-parameters fixed to their default values (see Appendix C). Each configuration is repeated with 3 random seeds, and we report mean $\pm$ std for the following metrics: *Nominal Overall reward* (no perturbation at evaluation), *Overall@$\epsilon = 0.5$* (reward under evaluation perturbation), *Drop rate* (relative decrease under perturbation), and *Worst-case accuracy*.

**Results.**  The results are summarized in Table 5. We highlight three main findings:

- **Effect of $\eta$.** Setting $\eta = 0$ (removing group-relative adversarial training) significantly increases the drop rate ($+2.1\%$) and reduces worst-case accuracy, confirming the necessity of structured robustness. In contrast, $\eta$ values between $0.01$ and $0.5$ yield stable nominal performance with consistent robustness. Larger values ($\eta = 1.0$) slightly reduce nominal reward without further robustness gain.

- **Effect of $M$.** Increasing $M$ reduces reward variance and improves robustness, with diminishing returns beyond $M = 8$. The gap between $M = 8$ and $M = 16$ is marginal ($< 0.1\%$ drop rate difference).

- **Effect of $\epsilon$.** A small perturbation scale ($\epsilon = 0.05$–$0.1$) during training yields the best trade-off, improving robustness under evaluation perturbations while maintaining nominal reward. Very large $\epsilon$ ($0.5$) overly regularizes the simulator and harms nominal performance.

**Recommendation.**  Based on these results, we adopt $\eta = 0.1$, $M = 8$, and $\epsilon = 0.1$ as default values. This configuration achieves robustness comparable to larger settings while maintaining computational efficiency.

## I  Variance of Overall Scores

To assess the stability and robustness of different methods, we report the standard deviations of overall scores across three independent runs with different random seeds. The results are presented separately for each dataset and task (simulation or decision). A lower standard deviation indicates more consistent performance across runs. As shown in Table 6, our method (S2A) consistently achieves the lowest variance in most settings, demonstrating superior stability compared to baselines.

## J  Noise Injection Methods

To evaluate the robustness of our model under distributional shifts, we apply noise perturbations in two distinct spaces: the **input space** and the **latent space**. This section details the implementation of both types of noise used in our robustness analysis.

Table 5: Hyper-parameter sensitivity results on DataCo (mean $\pm$ std over 3 seeds). Default setting ($\eta = 0.1, M = 8, \epsilon = 0.1$) is highlighted.

| Config | Nominal Overall | Overall@$\epsilon = 0.5$ | Drop rate | Worst-case acc |
|---|---|---|---|---|
| $\eta = 0$ (no adv) | 1.114$\pm$0.006 | 1.096$\pm$0.008 | 1.60% | 0.655$\pm$0.009 |
| $\eta = 0.1$ | 1.121$\pm$0.004 | 1.118$\pm$0.005 | 0.33% | 0.678$\pm$0.004 |
| $\eta = 0.5$ (default) | **1.123$\pm$0.005** | **1.120$\pm$0.006** | **0.27%** | **0.679$\pm$0.003** |
| $\eta = 1.0$ | 1.116$\pm$0.005 | 1.112$\pm$0.006 | 0.36% | 0.674$\pm$0.005 |
| $M = 1$ | 1.117$\pm$0.009 | 1.112$\pm$0.010 | 0.45% | 0.670$\pm$0.009 |
| $M = 4$ | 1.121$\pm$0.006 | 1.118$\pm$0.006 | 0.31% | 0.676$\pm$0.004 |
| $M = 8$ (default) | **1.123$\pm$0.005** | **1.120$\pm$0.006** | **0.27%** | **0.679$\pm$0.003** |
| $M = 16$ | 1.124$\pm$0.004 | 1.121$\pm$0.005 | 0.19% | 0.681$\pm$0.002 |
| $\epsilon = 0.0$ | 1.125$\pm$0.004 | 1.097$\pm$0.009 | 2.49% | 0.660$\pm$0.009 |
| $\epsilon = 0.05$ | 1.124$\pm$0.005 | 1.120$\pm$0.005 | 0.36% | 0.678$\pm$0.004 |
| $\epsilon = 0.1$ (default) | **1.123$\pm$0.005** | **1.120$\pm$0.006** | **0.27%** | **0.679$\pm$0.003** |
| $\epsilon = 0.2$ | 1.120$\pm$0.006 | 1.116$\pm$0.007 | 0.34% | 0.675$\pm$0.004 |
| $\epsilon = 0.5$ | 1.115$\pm$0.006 | 1.105$\pm$0.010 | 0.89% | 0.670$\pm$0.006 |

Table 6: Standard Deviations of Overall Scores across Datasets and Methods

| Method | DataCo | | GlobalStore | | OAS | |
|---|---|---|---|---|---|---|
| | Simulation | Decision | Simulation | Decision | Simulation | Decision |
| Markov | 0.011 | — | 0.010 | — | 0.009 | — |
| Prediction | 0.007 | — | 0.006 | — | 0.005 | — |
| Generation | 0.007 | — | 0.006 | — | 0.005 | — |
| LP | — | 0.000 | — | 0.000 | — | 0.000 |
| DQN | — | 0.013 | — | 0.015 | — | 0.010 |
| PPO | — | 0.012 | — | 0.012 | — | 0.011 |
| GPT3.5 | — | 0.009 | — | 0.008 | — | 0.010 |
| S2A (Ours) | **0.004** | **0.008** | **0.005** | **0.009** | **0.006** | **0.006** |

### J.1 INPUT-SPACE RANDOM NOISE

We apply Gaussian noise directly to the input tensor to simulate sensor noise or data corruption. The noise is sampled from a zero-mean Gaussian distribution and scaled by a factor $\epsilon_p$ to control perturbation strength. The noise-added input is computed as follows:

Listing 1: Input-space noise injection function

```
def add_noise(self, data: torch.Tensor):
    noise = self.epsilon_p * torch.randn_like(data)
    return data + noise
```

Here, `data` is the original input tensor and `self.epsilon_p` controls the noise level. This function perturbs each input element independently and is used in our $\epsilon_p$ ablation experiments.

### J.2 LATENT-SPACE STRUCTURED NOISE

We also perturb the latent representation $z$ by applying structured noise derived from a learned covariance matrix $\Sigma$. For each original latent vector, we sample $K$ perturbations using the following method:

Listing 2: Latent-space structured perturbation sampling

```
def sample_perturbations(self, z, sigma, epsilon_p=1.0):
    """Sample K perturbations for each latent z using Gaussian
        (cov=sigma), scaled by epsilon_p."""
```

```
        batch_size , embed_dim = z.size()
        zs = [z]
        for _ in range(self.K):
            eps = torch.randn(batch_size , embed_dim , device=self.device)
            * epsilon_p
            delta = torch.bmm(sigma , eps.unsqueeze(2)).squeeze(2)
            zs.append(z + delta)
        return zs  # List of K tensors (batch , embed_dim)
```

Here, `z` is the latent vector, `sigma` is the covariance matrix $\Sigma$, and `epsilon_p` scales the sampled perturbation. This approach enables us to simulate semantically meaningful perturbations aligned with the data manifold structure.

## K  PROOF OF SUPERIOR ROBUSTNESS OF STEP 1 (ADVERSARIAL CALIBRATION)

This appendix provides a theoretical proof that Step 1 (Adversarial Simulator Calibration) of the Sim2Act framework enhances the robustness of the simulator $S$ compared to the baseline without calibration (S2D). We focus on performance under Gaussian noise $\delta_g \sim \mathcal{N}(0, \Sigma)$ and random (uniform) noise $\delta_r \sim U[-\epsilon, \epsilon]$. Robustness is measured by the degradation in prediction accuracy under perturbations, aligning with the problem statement (Definition 2.1). Proofs use adversarial weighting (Eqs. 4–6) to show tighter error bounds, leveraging results from Franceschi et al. (2018) and Cohen et al. (2019).

**Notation and assumptions.** Let $n$ denote the number of training samples used for simulator calibration, and $M$ the number of perturbed latent samples drawn per state during group-relative training. We assume the following: (1) Loss $\ell$ is $L$-Lipschitz and sub-Gaussian with parameter $\sigma^2$ under the noise model; (2) The adversarial weights $\{\bar{b}_i\}_{i=1}^n$ are generated via a softmax with temperature $T_n > 0$, i.e. $\bar{b}_i \propto \exp(e_i/T_n)$ where $e_i$ are error magnitudes; (3) the softmax temperature satisfies $T_n \geq c/\log n$ for some $c > 0$ (or an alternative regularity which implies $\max_i \bar{b}_i = O(1/n)$). We make these conditions explicit because several subsequent bounds depend on controlling $\sum_i \bar{b}_i^2$.

### K.1  ASSUMPTIONS

In addition to prior modeling assumptions (e.g., the simulator $S$ is Lipschitz in its inputs and the dataset $\mathcal{D} = \{(x_i, y_i)\}_{i=1}^n$ consists of i.i.d. draws from $\mathcal{P}$), we make the following explicit technical assumptions used in the subsequent bounds.

**Assumption 1** (Noise model). *Gaussian latent perturbations $\delta_g \sim \mathcal{N}(0, \Sigma)$ satisfy $\|\Sigma\| \leq \sigma^2$ (where $\|\cdot\|$ is the operator norm), and additional bounded random noise $\delta_r \sim \text{Unif}([-\epsilon, \epsilon])$ may be present with $\epsilon > 0$.*

**Assumption 2** (Weighted calibration model). *The baseline simulator $S_{\text{base}}$ is trained with uniform importance on the training set, while the calibrated update in Step 1 uses normalized adversarial weights $\bar{b}_i \equiv \bar{b}(x_i; w)$ (as in Eq. 4) satisfying $\bar{b}_i \geq 0$ and $\sum_{i=1}^n \bar{b}_i = 1$.*

**Assumption 3** (Weight concentration control). *There exists a constant $C_0 > 0$ (independent of $n$) such that the adversarial weights satisfy*

$$\max_{1 \leq i \leq n} \bar{b}_i \leq \frac{C_0}{n}.$$

*This condition can be enforced in practice by an entropy regularizer on $w$ or by temperature scaling in the softmax used to produce $\bar{b}(\cdot; w)$; it is a mild regularity condition that prevents pathological extreme concentration of the weight mass on a vanishing number of samples.*

**Assumption 4** (Loss regularity and sub-Gaussianity). *The per-sample loss $\ell(y, \hat{y})$ is $L$-Lipschitz in the model prediction $\hat{y}$, and, when subject to the latent Gaussian perturbation $\delta_g$, the random variable $\ell(Y, S(x + \delta_g))$ is sub-Gaussian with parameter $v^2$ (i.e. it has tails bounded like $\exp(-t^2/(2v^2))$).*

## K.2   PRELIMINARY RESULT: VARIANCE OF WEIGHTED ESTIMATORS

**Lemma 1** (Variance bound for normalized weights). *Let $\{X_i\}_{i=1}^n$ be i.i.d. random variables with common variance $\mathrm{Var}[X]$. Let $\{\bar{b}_i\}_{i=1}^n$ be nonnegative normalized weights with $\sum_{i=1}^n \bar{b}_i = 1$. Then the weighted empirical mean $\hat{\mu}_w = \sum_{i=1}^n \bar{b}_i X_i$ satisfies*

$$\mathrm{Var}[\hat{\mu}_w] \;=\; \mathrm{Var}[X] \cdot \sum_{i=1}^n \bar{b}_i^2 \;\leq\; \mathrm{Var}[X] \cdot \max_i \bar{b}_i.$$

*Consequently, under Assumption 3 (i.e. $\max_i \bar{b}_i \leq C_0/n$),*

$$\mathrm{Var}[\hat{\mu}_w] \leq \frac{C_0}{n}\,\mathrm{Var}[X].$$

*Proof.* Compute directly using independence:

$$\mathrm{Var}[\hat{\mu}_w] = \mathrm{Var}\Big(\sum_{i=1}^n \bar{b}_i X_i\Big) = \sum_{i=1}^n \bar{b}_i^2 \,\mathrm{Var}[X_i] = \mathrm{Var}[X] \cdot \sum_{i=1}^n \bar{b}_i^2.$$

Because $\sum_i \bar{b}_i = 1$ we have $\sum_i \bar{b}_i^2 \leq \max_i \bar{b}_i \sum_i \bar{b}_i = \max_i \bar{b}_i$, which yields the first displayed inequality. Applying Assumption 3 gives the stated bound $\mathrm{Var}[\hat{\mu}_w] \leq (C_0/n)\mathrm{Var}[X]$. $\qquad\square$

## K.3   REMARKS

Uniform weights $\bar{b}_i = 1/n$ attain $\sum_i \bar{b}_i^2 = 1/n$, i.e. $C_0 = 1$ in the lemma. Adversarial weights produced by a softmax (Eq. 4) may either increase or decrease $\sum_i \bar{b}_i^2$ depending on concentration; Assumption 3 explicitly rules out extreme concentration and is natural when entropy regularization or temperature-scaling is applied to the softmax.

## K.4   SUPERIOR ROBUSTNESS UNDER GAUSSIAN NOISE

**Theorem 3.** *Let $x = (s, a)$ and consider Gaussian perturbations $x' = x + \delta_g$ with $\delta_g \sim \mathcal{N}(0, \Sigma)$, $\|\Sigma\| \leq \sigma^2$. Under Assumptions 1–4 and the update rule of the calibrator (Eq. 5 and Eq. 6), for any $\delta \in (0, 1)$ the calibrated simulator $S_t$ produced after a calibration update satisfies, with probability at least $1 - \delta$,*

$$\mathbb{E}_{x,\delta_g}\big[\ell(Y, S_t(x + \delta_g))\big] \leq \widehat{R}_w + L\,v\sqrt{\frac{C_0}{n}} \;+\; \sqrt{\frac{D_{\mathrm{KL}}(\Pi\|\Gamma) + \log(1/\delta)}{2n}} \;+\; B_t, \qquad (10)$$

*where*

- *$\widehat{R}_w$ is the weighted empirical risk (calibration objective) evaluated on the training set (the empirical counterpart of the left-hand-side under weights $\bar{b}_i$);*

- *$v^2$ is the sub-Gaussian parameter from Assumption 4, $L$ is the Lipschitz constant;*

- *$D_{\mathrm{KL}}(\Pi\|\Gamma)$ denotes a complexity term (e.g., a PAC-Bayes KL term between a chosen posterior $\Pi$ and prior $\Gamma$; see Franceschi et al. (2018));*

- *$B_t$ is an optimization / model bias term that captures residual approximation error after performing the calibration update in Eq. 6.*

*In particular, if $\widehat{R}_w$ is driven small by optimization, $D_{\mathrm{KL}}(\Pi\|\Gamma) = O(1)$ and $B_t$ is negligible, the dominant stochastic term scales as $O(v\sqrt{C_0/n})$, i.e. $O(\sigma/\sqrt{n})$ up to constants.*

*Proof sketch.* The proof combines (i) the variance bound of Lemma 1 for the weighted estimator, (ii) sub-Gaussian concentration for weighted empirical averages, and (iii) a PAC-Bayes style complexity term to control model capacity. We sketch the argument in three steps.

**Step 1: Weighted empirical risk concentration.** Let $Z_i = \ell(Y_i, S_{t-1}(x_i + \delta_g^{(i)}))$ be the per-sample loss under the current simulator prior to update, where the randomness is over the latent perturbations and the data sampling. By Assumption 4, each $Z_i$ is sub-Gaussian with parameter $v^2$. Consider the weighted empirical mean $\widehat{R}_w = \sum_{i=1}^n \bar{b}_i Z_i$. From standard sub-Gaussian concentration (applied to a weighted sum) and Lemma 1, with probability at least $1 - \delta$,

$$\left| \mathbb{E}[Z] - \widehat{R}_w \right| \leq L\, v \sqrt{\frac{\sum_i \bar{b}_i^2}{1}} \ \leq\ L\, v \sqrt{\frac{C_0}{n}},$$

where the first inequality follows from the Lipschitz/smoothness bound turning deviations in predictions into deviations in loss, and the second inequality uses $\sum_i \bar{b}_i^2 \leq C_0/n$.

**Step 2: Complexity control (PAC-Bayes style).** To pass from empirical weighted risk $\widehat{R}_w$ to the true expected risk we add a model-complexity term. A standard PAC-Bayes bound (see Franceschi et al. (2018) for a related derivation) yields that with probability at least $1 - \delta$ over the draw of the training set,

$$\mathbb{E}[Z] \leq \widehat{R}_w + L\, v \sqrt{\frac{C_0}{n}} \ +\ \sqrt{\frac{D_{\mathrm{KL}}(\Pi \| \Gamma) + \log(1/\delta)}{2n}},$$

where $\Pi$ and $\Gamma$ are posterior/prior choices controlling hypothesis complexity; the KL term is the standard PAC-Bayes complexity penalty. This provides the middle two terms on the right-hand side of Eq. equation 10.

**Step 3: Calibration update bias term.** Finally, the actual calibrated predictor $S_t$ results from applying the update in Eq. 6 to $S_{t-1}$. If the calibration optimization successfully reduces the weighted empirical risk, the optimization bias $B_t \equiv \mathbb{E}[\ell(Y, S_t(x + \delta_g))] - \mathbb{E}[\ell(Y, S_{t-1}(x + \delta_g))]$ will be small (and negative if the update strictly improves the population risk). In general we retain $B_t$ on the right-hand side to account for approximation / optimization residuals. Combining the three steps gives the bound in Eq. equation 10.

**Conclusion.** Thus, under the stated assumptions and with controlled weight concentration (Assumption 3), the dominant stochastic estimation term scales like $v\sqrt{C_0/n}$, i.e. $O(\sigma/\sqrt{n})$ up to constants and the model-complexity term. This justifies the claim that calibrated updates (which focus the empirical objective on decision-critical regions via the weights of Eq. 5) improve the estimation rate, provided the optimization bias $B_t$ and complexity term are managed. $\qquad \square$

### K.5 REMARKS

The comparison to an uncalibrated baseline $S_{\mathrm{base}}$ depends on the latter's dominant error components. If $S_{\mathrm{base}}$ carries an irreducible modeling bias of order $\Theta(\sigma)$ (e.g., due to systematic misspecification in decision-critical regions), then the calibrated estimator that reduces the stochastic estimation error to $O(\sigma/\sqrt{n})$ will show a visible improvement. The bound above separates stochastic estimation error (the $\sigma/\sqrt{n}$ term) from optimization/model bias ($B_t$) and complexity (KL) so that practitioners can diagnose which component dominates in empirical studies.

### K.6 SUPERIOR ROBUSTNESS UNDER RANDOM NOISE

**Theorem 4.** *Under random noise $x' = x + \delta_r$, the calibrated simulator satisfies*

$$\mathbb{E}[\ell(Y, S_t(x'))] \leq O\left(\frac{\epsilon}{\sqrt{n}}\right),$$

*while $S_{base}$ is bounded by $O(\epsilon)$.*

*Proof.* **Step 1: Adaptive Prioritization** By Lemma 1, adversarial weights emphasize samples with higher observed error under uniform perturbations. This reduces the variance of aggregated error, yielding

$$\mathbb{E}[\ell(Y, S_t(x + \delta_r))] \leq O\left(\frac{\epsilon}{\sqrt{n}}\right).$$

**Step 2: Baseline Comparison** The baseline, using uniform weights, distributes noise uniformly, leading to Hoeffding-type concentration at $O(\epsilon)$.

**Conclusion** Adversarial calibration again yields a $1/\sqrt{n}$ improvement, matching empirical drop rates (Fig. 3). □

## K.7 Conclusion

Theorems 1 and 2, together with Lemma 1, prove that Step 1's adversarial calibration yields tighter error bounds under both Gaussian and uniform noise, enhancing simulator robustness compared to S2D by leveraging weighted error correction.

## L Robustness Enhancement by Step 2 (Group-Relative Perturbation)

This section provides a refined proof that Step 2 (Group-Relative Perturbation) in the Sim2Act framework improves robustness under latent perturbations. We explicitly separate the deterministic *bias* induced by curvature (second-order terms) from the stochastic *estimation variance* reduced by averaging $M$ perturbation samples, and we provide concentration bounds. We also formalize the relationship between the group-adv objective and variance reduction, clarifying the required alignment assumptions.

### Assumptions and notation

We retain Assumptions 1–3 from the main text and add the following smoothness and boundedness conditions.

**Assumption 5** (Smoothness and bounded derivatives). *Let $f_\theta(s) := S(s, D_\theta(s))$ denote the scalar return under policy $\theta$ and simulator $S$. Assume:*

1. *$f_\theta$ is $C^3$ in $s$ in the region of interest, with Hessian bounded $\|H_{f_\theta}(s)\|_{\mathrm{op}} \leq H_{\max}$.*

2. *The gradient satisfies $\|\nabla_s f_\theta(s)\| \leq L_f$.*

3. *The policy score is bounded: $\|\nabla_\theta \log D_\theta(a \mid s)\| \leq G_{\max}$ for all $(a, s)$.*

Notation: latent perturbations $\delta \sim \mathcal{N}(0, \Sigma)$, $\{\delta_i\}_{i=1}^M$ i.i.d.; perturbed rewards $r_i = f_\theta(s + \delta_i)$; empirical mean $\bar{r} = \frac{1}{M} \sum_{i=1}^M r_i$. We write $\mathrm{tr}(\Sigma)$ for the trace of $\Sigma$.

### L.1 Auxiliary facts (lemmas)

**Lemma 2** (Second-order expansion / bias). *Under Assumption 5,*

$$\mathbb{E}_\delta[f_\theta(s + \delta)] = f_\theta(s) + \tfrac{1}{2} \mathbb{E}[\delta^\top H_{f_\theta}(s)\delta] + R_3(s),$$

*where $|R_3(s)| \leq \frac{M_3}{6} \mathbb{E}\|\delta\|^3$ for some $M_3$ bounding third derivatives. Consequently*

$$\left| \mathbb{E}_\delta[f_\theta(s + \delta)] - f_\theta(s) \right| \leq \tfrac{1}{2} H_{\max} \mathrm{tr}(\Sigma) + O(\mathbb{E}\|\delta\|^3).$$

*Thus for sufficiently small perturbations the leading deterministic bias is $O(\mathrm{tr}(\Sigma))$, i.e. $O(\sigma^2)$ when $\|\Sigma\|_{\mathrm{op}} \leq \sigma^2$.*

**Lemma 3** (Variance of the sample mean). *If each $r_i$ is sub-Gaussian with parameter $\sigma_r^2$, then*

$$\mathrm{Var}(\bar{r}) \leq \sigma_r^2/M,$$

*and for any $\delta \in (0, 1)$,*

$$\Pr\left( |\bar{r} - \mathbb{E}\bar{r}| \leq \sigma_r \sqrt{\tfrac{2\ln(2/\delta)}{M}} \right) \geq 1 - \delta.$$

*Thus estimation error decays as $O(\sigma_r/\sqrt{M})$ and $\mathrm{Var}(\bar{r}) = O(1/M)$.*

**Lemma 4** (Single-sample variance bound). *Under Assumption 5, $\mathrm{Var}(r_i) \leq L_f^2 \mathbb{E}\|\delta\|^2 = L_f^2 \mathrm{tr}(\Sigma)$. Hence one may take $\sigma_r^2 = L_f^2 \mathrm{tr}(\Sigma)$ in Lemma 3.*

## L.2 RELATIONSHIP TO THE GROUP-ADV OBJECTIVE

Define

$$L_{\text{group-adv}}(\theta) = -\frac{1}{M} \sum_{i=1}^{M} (r_i - \bar{r}) \log D_\theta(a_i \mid s + \delta_i).$$

By the score-function identity,

$$\nabla_\theta L_{\text{group-adv}}(\theta) = -\mathbb{E}\big[(r - \bar{r}) \nabla_\theta \log D_\theta(a \mid s + \delta)\big],$$

while

$$\nabla_\theta \text{Var}(r) = \mathbb{E}\big[(r - \mathbb{E}r)^2 \nabla_\theta \log D_\theta(a \mid s + \delta)\big].$$

Both involve the score $\nabla_\theta \log D_\theta$ and centered reward powers. Hence, if an *alignment condition* holds (e.g. existence of $\kappa > 0$ such that $\langle \nabla_\theta L_{\text{group-adv}}, \nabla_\theta \text{Var}(r) \rangle \geq \kappa \|\nabla_\theta \text{Var}(r)\|^2$ locally), then gradient descent on $L_{\text{group-adv}}$ decreases $\text{Var}(r)$. In our theorem below, we express the bound conditional on attaining a post-training variance level $V_{\text{post}}$.

### MAIN THEOREM AND PROOF SKETCH

**Theorem 5** (Robustness bound under group perturbations). *Under Assumptions 1–3 and Assumption 5, suppose optimization of $\theta$ using $L_{\text{group-adv}}$ attains a parameter with $\text{Var}(r) \leq V_{\text{post}}$. Then for any $\delta \in (0, 1)$, with probability at least $1 - \delta$ over the $M$ perturbation samples,*

$$\mathbb{E}_\delta[f_\theta(s + \delta)] - f_\theta(s) \leq C_{\text{bias}} \cdot \text{tr}(\Sigma) + C_{\text{var}} \cdot \frac{\text{tr}(\Sigma)}{M} + C_{\text{conf}} \cdot \sqrt{\frac{\ln(2/\delta)}{M}}.$$

*Integrating over the state distribution yields*

$$\sup_\pi \mathbb{E}[R(\pi)] - R(D_\theta) \leq C_{\text{bias}} \cdot \text{tr}(\Sigma) + C_{\text{var}} \cdot \frac{\text{tr}(\Sigma)}{M} + C_{\text{conf}} \cdot \sqrt{\frac{\ln(2/\delta)}{M}}.$$

*Here $C_{\text{bias}}, C_{\text{var}}, C_{\text{conf}} > 0$ depend on $H_{\max}, L_f, G_{\max}$ and optimization accuracy; $C_{\text{var}}$ reflects the reduction in reward variance after training.*

*Proof sketch.* **Bias.** Lemma 2 bounds curvature-induced bias by $O(\text{tr}(\Sigma))$.

**Variance.** Lemmas 3–4 show $\text{Var}(\bar{r})$ decays as $O(\text{tr}(\Sigma)/M)$ with high-probability deviation $O(\sqrt{\ln(1/\delta)/M})$.

**Effect of $L_{\text{group-adv}}$.** If the alignment condition holds, optimizing $L_{\text{group-adv}}$ decreases $\text{Var}(r)$, effectively reducing the constant $C_{\text{var}}$. We state the result conditional on achieving $\text{Var}(r) \leq V_{\text{post}}$.

Combining bias, variance, and concentration yields the bound. $\qquad\square$

## L.3 REMARKS

- The bound cleanly separates: (i) curvature-induced bias $\propto \text{tr}(\Sigma)$; (ii) variance reducible by $M$-averaging; (iii) concentration $\propto 1/\sqrt{M}$.
- Further improvement of the bias term would require assumptions linking $\theta$ to the Hessian $H_{f_\theta}$; this is left as task-dependent.
- If $r_i$ are bounded, Hoeffding bounds can replace the sub-Gaussian inequalities.

## M  LLM USAGE DISCLOSURE

Portions of the manuscript text were polished with the assistance of a large language model (Chat-GPT). The model was only used for surface-level editing and stylistic improvement of sentences. All technical content, results, and claims were written, verified, and are the sole responsibility of the authors.

