# OpenReview forum: "Sim2Act: Robust Simulation-to-Decision Learning via Adversarial Calibration and Group-relative Perturbation"
_ICLR.cc/2026/Conference — ICLR 2026 Conference Withdrawn Submission_

### Official Review · Reviewer_1iRH · 2025-10-30

**Soundness:** 1
**Presentation:** 3
**Contribution:** 1
**Rating:** 2
**Confidence:** 5

**Summary:**

This paper addresses the disparity between simulation and real datasets (i.e. sim2real) where shifts in state distributions create uncertainty and performance degradation using adversarial simulator calibration and a group-relative perturbation mechanism for robust policies. The learned simulator perturbs states where prediction errors are most harmful to the agent, and the agent is optimized under the simulator using a group-relative regret. The experiments show that the proposed method is robust to distribution shifts in logistics data as compared to baselines.

**Strengths:**

- The paper provides an interesting insight into group-relative advantage and its role in robustness and stabilizing policy gradient updates.
- The latent-space perturbations used in the adversarial simulator and group-relative perturbations are an interesting idea. It seems reasonable that this way of viewing perturbations would produce more semantically meaningful perturbations with respect to the task.

**Weaknesses:**

#### Notation
- The paper appears to use reinforcement learning (RL) terminology and ideas (i.e., advantage, state/action distributions), but does not define or discuss them explicitly. Doing so (for instance, by discussing the underlying MDP of the simulator and dataset) would likely yield more thorough theoretical results and greatly aid in mapping this work to prior work.

#### Experiments
- The table captions should be more descriptive. Particularly, the metrics in Table 1 are not explained.
- There are no robust baseline methods present, which are important in assessing the effectiveness of robustness work. GPT 3.5 is shown, but it is not clear how or why it is used to solve the tasks.
- The proposed method underperforms the preceding method (S2D) in the "Status" metric in Table 1, but it is not discussed why or what the metric means.

#### Methodology
- The adversarial calibrator is stated to be inspired by or taken from preceding work [1]. It should be discussed if this is a direct adaptation or a novel iteration, and what challenges present in the original method are addressed in this work.
- While interesting, the latent-space perturbations do not have a stated advantage or motivation as compared to state perturbations such as PGD [2], which are a well-defined setting in adversarial literature [3].
- Equation 8 appears to be a regret term similar to that in adversarial RL [4] that is either counterintuitive or incorrect. Regret terms are minimized, since an effort to maximize regret would result in the minimization of the target reward $\bar{r}$. When minimizing the loss in Equation 9, under a negative $\mathcal{S}(s,a)$, Equation 8 would be maximized unless the regret is negated as well.



[1] Haoyue Bai, Haoyu Wang, Nanxu Gong, Xinyuan Wang, Wangyang Ying, Haifeng Chen, Yanjie Fu: Supply Chain Optimization via Generative Simulation and Iterative Decision Policies. CoRR abs/2507.07355 (2025)

[2] Aleksander Madry, Aleksandar Makelov, Ludwig Schmidt, Dimitris Tsipras, Adrian Vladu: Towards Deep Learning Models Resistant to Adversarial Attacks. ICLR (Poster) 2018
.
[3] Huan Zhang, Hongge Chen, Chaowei Xiao, Bo Li, Mingyan Liu, Duane S. Boning, Cho-Jui Hsieh: Robust Deep Reinforcement Learning against Adversarial Perturbations on State Observations. NeurIPS 2020

[4] Roman Belaire, Arunesh Sinha, Pradeep Varakantham: On Minimizing Adversarial Counterfactual Error in Adversarial Reinforcement Learning. ICLR 2025

**Questions:**

- Is the adversarial calibrator different from the cited S2D?
- Why is GPT 3.5 used as a baseline solver for the task?
- What is the motivation for using latent perturbations and an encoder-decoder setup?
- Can the method be adapted to work under state perturbations, i.e. PGD?

---

### Official Review · Reviewer_91xm · 2025-10-30

**Soundness:** 2
**Presentation:** 2
**Contribution:** 2
**Rating:** 4
**Confidence:** 4

**Summary:**

The paper introduces Sim2Act (S2A), a two-stage framework for achieving robust simulation-to-decision (Sim2Dec) learning in digital twins, particularly suited to mission-critical domains such as supply chains and power systems. Sim2Act enhances both components of the pipeline: (1) it improves simulation fidelity through adversarial calibration, and (2) strengthens policy robustness via group-relative perturbations. The overarching goal is to realize non-disruptive robustness, ensuring stable performance under perturbations while preserving decision quality.

**Strengths:**

1. The paper grounds the Sim2Act framework in practical, high-stakes domains such as supply chains, power grids, and robotics, where inherent noise, uncertainty, and the cost/risk of real-world interaction pose significant challenges. This clear and realistic context enhances the work's credibility and applied relevance.

2. The experimental validation is robust and well-structured, utilizing three distinct real-world supply chain datasets: DataCo, GlobalStore, and OAS. The results consistently demonstrate quantitative gains supporting the paper's claims.

**Weaknesses:**

1. Limited Experimental Scope: The paper claims broad applicability to high-stakes domains like robotics and power grids, but all experiments are confined to discrete-action, logistics-focused supply chain datasets. The generalizability of the method to complex, high-dimensional continuous control problems or systems with non-stationary dynamics remains unproven.

2. Lack of Qualitative Policy Analysis: While the quantitative results are strong, the evaluation lacks depth in analyzing policy behavior. The paper doesn't explore how the calibrated simulator fundamentally changes specific decision trajectories, which actions are avoided, or whether the policy's failure modes are genuinely safer or more interpretable.

3. Notation Issues : (1) The main problem statement (Equation 2) is **ill-typed** because the simulator $\mathcal{S}(s_t, a_t)$ is formally defined to output a **tuple** of the next state and reward $(\hat{s}_{t+1}, \hat{r}_t)$ in line 246. However, the objective function sums $\mathcal{S}(\cdot)$ over time, implying it must be a scalar reward. (2) The parameter $\mathbf{w}$ is introduced in the Method Overview (Section 3.1) in the calibrator notation $\overline{b}(s,a,w)$ in line 162, but it is not formally defined at that point.

**Questions:**

1. How reliable is the learned covariance estimator $\mathbf{\Sigma(s,a)}$ in practice? For instance, does its estimated variance remain stable across different random seeds? Additionally, evaluating only three seeds might be too limited to assess robustness in reinforcement learning settings.

2. The Group-Relative Perturbation technique, by aggregating results over multiple perturbed states, shares a functional similarity with ensemble methods. Would adopting an explicit ensemble approach for the simulator (e.g., training multiple $\mathcal{S}$ models) yield a superior or more robust estimate of the latent uncertainty $\mathbf{\Sigma(s, a)}$ compared to relying on a single simulator's variance output?

3. How does Sim2Act scale to continuous control domains (e.g., MuJoCo or D4RL tasks)? Does the adversarial calibration bottleneck model performance for large state spaces?

---

### Official Review · Reviewer_poEg · 2025-10-31

**Soundness:** 3
**Presentation:** 2
**Contribution:** 3
**Rating:** 2
**Confidence:** 4

**Summary:**

This paper presents Sim2Act, a two-step framework designed to improve robustness in simulation-to-decision (Sim2Dec) learning. The authors aim to mitigate two central issues: biased surrogate simulators and fragile decision policies under distributional shifts. Their approach combines (1) adversarial calibration to enhance simulator fidelity in decision-critical regions, and (2) group-relative perturbation to improve policy robustness while maintaining performance stability. Experiments are conducted on both synthetic and real-world datasets (DataCo, GlobalStore, and OAS), demonstrating the proposed method’s ability to handle noisy and biased data.

**Strengths:**

The paper addresses an important problem of robustness in Sim2Dec learning, which is crucial for digital twin applications and model-based RL under noisy or biased environments.

The idea of emphasizing simulator errors in decision-critical regions is intuitively meaningful and could inspire future work in coupling model fidelity and policy robustness.

The paper includes both synthetic and real-world experiments (DataCo, GlobalStore, and OAS), which help demonstrate practical applicability.

**Weaknesses:**

Bridging the gap between simulation accuracy and decision robustness is important for digital twin applications. However, the novelty and empirical strength of the proposed approach appear limited. The adversarial calibration component, while conceptually sound, resembles the mechanism used in Sim2Dec, and the distinction between the two frameworks is not clearly articulated. Section 3.2 in particular reads as a close variant of Sim2Dec’s adversarial training procedure. From the experimental results, Sim2Act shows only marginal improvement over Sim2Dec, suggesting that the contribution is incremental rather than substantially novel.

The simulator should focus on regions where prediction errors most affect policy actions, which is intuitive and potentially useful. However, the current design, which trains the simulator and policy in separate stages, weakens this argument. The authors assume that prediction errors naturally concentrate in decision-critical areas, yet without joint training, there is no guarantee that simulator inaccuracies indeed lead to different actions. In fact, prediction errors may or may not cause the policy to output distinct decisions, depending on the policy’s sensitivity in specific state regions. Therefore, the crucial factor is whether the policy is sensitive to simulator errors in those states, not merely whether the simulator is inaccurate in those states. The cited reference at line 179 mentions that inaccuracies can compound and degrade policy performance, but it does not support the claim that policy sensitivity aligns with the distribution of simulator prediction errors.

Equation (5) is difficult to interpret. The paper states that the method increases weights for inaccurate $(s,a)$ pairs, yet the weighting term
$\bar{b}(s, a;w)$ is defined in an indirect way. It is unclear why this formulation is preferred over more straightforward functions such as squared or exponential error. Moreover, since $\bar{b}$ depends primarily on the inner product $<s,w_a>$, where $w_a$ is a learnable vector for each action, the relationship between this design and actual prediction error remains ambiguous.

Another issue arises from the claim that the method works well on noisy datasets. Typically, noisy samples correspond to larger prediction errors, and increasing their weights could amplify noise instead of improving robustness. The paper would benefit from a more thorough analysis or ablation study to demonstrate that the model does not overfit to noise-dominant regions.

Section 3.2.2 is also somewhat confusing. The authors introduce stochasticity on the policy side rather than modeling uncertainty in the environment, but it is not clear why this design captures robustness more effectively. Additionally, in Equation (9), when the reference reward is absent, the authors set $r^∗=0$, which effectively drives $S(s, a)$ toward zero. However, “no reference reward” does not necessarily imply a zero-reward scenario, and this simplification may bias the optimization process. Finally, the description of how ChatGPT-3.5 is used as a policy is vague and needs more concrete explanation.

**Questions:**

- How is Sim2Act fundamentally different from Sim2Dec in terms of both formulation and training procedure?

- Without joint training of the simulator and policy, how can the method ensure that prediction errors truly affect policy actions, rather than occurring in regions irrelevant to decision-making?

- In Equation (5), why not directly use a squared or exponential loss instead of the indirect weighting function $\bar{b}(s, a;w)$?

- Noisy samples usually have larger prediction errors. Wouldn’t increasing their weights risk amplifying noise and harming generalization?

- How exactly is ChatGPT-3.5 used as a policy in your framework? What role does it play in the training or evaluation process?

---

### Official Review · Reviewer_wZnc · 2025-10-31

**Soundness:** 3
**Presentation:** 3
**Contribution:** 2
**Rating:** 4
**Confidence:** 2

**Summary:**

The paper proposes SIM2ACT, a two-stage sim-to-decision pipeline: (1) adversarial simulator calibration that reweights errors in decision-critical regions and applies a closed-form correction; (2) group-relative decision training that perturbs the simulator’s latent state, then updates the policy with a GRPO-style group-mean–centered objective to favor actions that remain good across the sampled neighborhood. Experiments on supply-chain datasets claim improved robustness metrics and comparable nominal performance, with ablations for each stage.

**Strengths:**

- Decision-focused calibration is well motivated: it aims at reducing errors that matter for downstream decisions rather than global fit.
- The group-relative objective is simple and critic-free, echoing GRPO ideas where a group average serves as the baseline.

**Weaknesses:**

- Positioning vs robust RL is loose. The method is not a two-player min–max robust MDP or RARL; it optimizes relative performance over a sampled ensemble. Direct comparisons to robust MDPs and RARL/EPOpt are missing.
- Uncertainty modeling is narrow: latent Gaussian noise from the encoder. There is no analysis of distributional misspecification (heavy tails, multimodality) or sensitivity to the number/scale of perturbations
- Robustness baselines and metrics under true worst-case or tail risk (CVaR/DRO) are abscent

**Questions:**

- Could your latent-perturbation neighborhood be replaced or augmented with multimodal samplers such as diffusion planners or flow-matching paths to better capture diverse futures? Any preliminary results with Diffuser-style trajectory denoising or Flow Matching-driven latent paths would be valuable.
- Can you report min-over-group or CVaR@α returns and compare against robust MDP baselines to clarify the robustness claim?
- How sensitive are results to the latent covariance source and to the number of samples M? Any failures with heavy-tailed noise?

---

### Note · Authors · 2026-01-23

I have read and agree with the venue's withdrawal policy on behalf of myself and my co-authors.